# Evaluation of Three Gridded Precipitation Products in Characterizing Extreme Precipitation over the Hengduan Mountains Region in China

**Wenchang Dong, Genxu Wang, Li Guo** 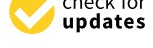 **, Juying Sun and Xiangyang Sun \***

State Key Laboratory of Hydraulics and Mountain River Engineering, College of Water Resource and Hydropower, Sichuan University, Chengdu 610065, China
\* Correspondence: sunxy@scu.edu.cn

**Abstract:** Extreme precipitation events can lead to severe mountain hazards, and they have therefore received widespread attention. The study of extreme precipitation can be hindered by the insufficient number and uneven distribution of rain gauge stations, especially in mountainous areas with complex terrain. In this study, the daily precipitation data of three gridded precipitation products (Integrated Multi-satellite Retrievals for GPM, IMERG; Multi-Source Weighted-Ensemble Precipitation, MSWEP; and Tropical Rainfall Measuring Mission, TRMM) were compared with rain gauge observations at 62 ground stations from 2001 to 2016 over the Hengduan Mountain region in China. Deviations between the gridded and ground precipitation datasets were compared using four daily heavy rainfall sequences. Various extreme precipitation indices were used to evaluate the performance of selected precipitation products. The results show that IMERG and TRMM are better than MSWEP in characterizing extreme precipitation. The accuracy of these three products in detecting heavy precipitation varied with altitude gradient. All products provided more accurate estimates of heavy precipitation in higher-altitude areas than in lower-altitude areas. Notably, they are more applicable for heavy precipitation detection in subalpine or alpine regions, and there are still uncertainties in capturing the accurate characterization of extreme precipitation at low (<1000 m) altitudes in the Hengduan Mountain region. These precipitation products should be used with caution in future applications when analyzing extreme precipitation at low elevations.

**Keywords:** extreme precipitation; altitude gradient; Hengduan Mountains region; IMERG; TRMM

## 1. Introduction

Extreme precipitation is the amount of precipitation that exceeds a certain threshold [1], and it can trigger a series of natural disasters that can cause severe damage to human life and the natural environment [2,3]. With global climate change, the accelerated rate of the water cycle has caused striking variation in precipitation frequency distributions [4,5]. Consequently, extreme rainstorm events are becoming more intensive [6,7], leading to frequent landslides, mudslides, and other mountain disasters [8,9]. The Hengduan Mountains region (HMR) in China is a mountainous, hazard-prone area [10]. Extreme precipitation is the main trigger for mountain hazards [11], the incidence of which is increasing in the HMR [12–14]. However, achieving finer spatial and temporal resolution observations for extreme precipitation analysis in mountainous regions is challenging, because of the sparse and uneven spatial distribution of ground observation stations [15,16]. Gridded precipitation products, which are mainly based on passive microwave (PMW) and calibrated infrared (IR), have the advantages of high spatial and temporal resolutions and wide coverage [17]. Therefore, there is a pressing need to improve the understanding of extreme precipitation events using gridded precipitation products in the HMR, and assessing the applicability of gridded precipitation products is a prerequisite for product selection.

The spatiotemporal variation in precipitation is closely related to the topography [18]. Elevation gradients of precipitation are common in mountainous areas [19,20]. On a regional scale, the distribution of heavy rainfall presents a significant pattern of spatial heterogeneity owing to the complex topography of mountainous regions [21,22]. On a larger spatial scale, airflow rises under the action of uplifted terrain and, consequently, water molecules converge, condense, and form a precipitation peak [23]. Guo et al. (2020) found a significant relationship between extreme precipitation indices and altitude in the HMR [24]. The results of spatial regression models in Taiwan suggest that terrain plays a primary role in extreme precipitation fluctuations [25]. Hence, the effects of elevation cannot be ignored while studying extreme precipitation over mountainous areas. Accordingly, the applicability of various gridded products in HMR should consider the performance of extreme precipitation at different elevations.

The most popular grid precipitation products include the Climate Hazards Group Infrared Precipitation with Station data (CHIRPS) [26], Multi-Source Weighted-Ensemble Precipitation (MSWEP) [27], Tropical Rainfall Measuring Mission (TRMM) Multi-satellite Precipitation Analysis (TMPA) [28], and global precipitation measurement (GPM) [29]. Unfortunately, the uncertainties caused by systematic errors inherent in satellite datasets limit the direct application of satellite-based precipitation products [30]. Satellite-based data resources, inversion algorithms, gauge procedures, and sensor performance can all affect the accuracy of precipitation product datasets [31]. To date, many studies have investigated the characteristics and applicability of precipitation product datasets from various aspects [32,33]. For example, MSWEP V2 is the first fully global precipitation dataset with a $0.1°$ resolution, which exhibits realistic spatial patterns in mean, magnitude, and frequency for global precipitation [27]. Therefore, among the available gridded precipitation products, MSWEP is considered more reliable for drought forecasting and suitable for studying long-term mesoscale precipitation [34–37]. Prior studies have directly compared TRMM and Integrated Multi-satellite Retrievals for GPM (IMERG) products and quantified their performance and bias. For example, the GPM sensors of IMERG can detect light (<0.5 mm/h) and solid precipitation more accurately than those of TRMM; therefore, the IMERG dataset is considered superior to the TRMM dataset [38], whereas TRMM exhibits superior detection performance at large time and space scales to IMERG [39,40]. Luo et al. (2021) proposed a new method for assessing precipitation products in ungauged basins or areas with limited ground-based precipitation monitoring [41]. These results confirmed the relatively good accuracy of these three widely used products and provided the basis for selecting the appropriate one; however, there are few clues regarding the accuracy and applicability of precipitation products to detect extreme precipitation at different elevations in mountainous regions.

Some studies examined the ability of TRMM to identify extreme precipitation [42,43]. For example, TRMM 3B42-V7 can reproduce extreme indices better in Eastern China than in Western China [43]. Liu et al. (2020) suggested that GPM-based estimates are useful for simulating extreme precipitation events in the Yangtze River Basin [44]. The ability of these popular precipitation products to detect extreme precipitation (especially high-intensity precipitation that causes flooding) lacks an accurate comparison. Furthermore, although changes in extreme precipitation events show spatial variability [45,46], previous studies rarely mentioned the corresponding topographic features [47,48]. Overall, an in-depth comparison of multiple gridded precipitation datasets is required to further explore the reasons for the relative performances of different datasets in detecting extreme precipitation in mountainous regions.

This study quantified the performance of various products in capturing the characteristics of extreme precipitation and evaluated their applicability and accuracy in a steep mountainous region. We compared the latest versions of the MSWEP, TRMM, and IMERG gridded precipitation products with ground observations using data from 2001 to 2016. This study provides a guide for the potential use of precipitation products for future disaster prevention and risk management in mountainous regions.

## 2. Materials and Methods

### 2.1. Study Area

The HMR (24°35′–33°34′N, 96°56′–104°30′E) is located in Southwest China (Figure 1). It lies at the junction of the Qinghai–Tibet Plateau, Yunnan–Guizhou Plateau, and Sichuan Basin [49], covering an area of approximately $5 \times 10^5$ km$^2$. It is also the transition zone between the first and second terrains of China's topography, which creates complex terrain conditions, and the regional elevation ranges from 330 to 6400 m, decreasing from the northwest to the southeast [13]. The maximum elevation difference in the area can exceed 6000 m, with the characteristics of high mountains, deep valleys, and large relative elevation differences. The HMR is located in a typical monsoon climate region, controlled by the South Asian and East Asian monsoons. In addition, it is influenced by the Tibetan Plateau monsoon and westerlies [8]. The complex terrain and various monsoons cause significant spatial and temporal differences in precipitation [50]. The average annual rainfall is as high as 1137 mm, and approximately 75–90% of the annual rainfall is concentrated from June to September [14,21], thus leading to frequent disasters in the mountains.

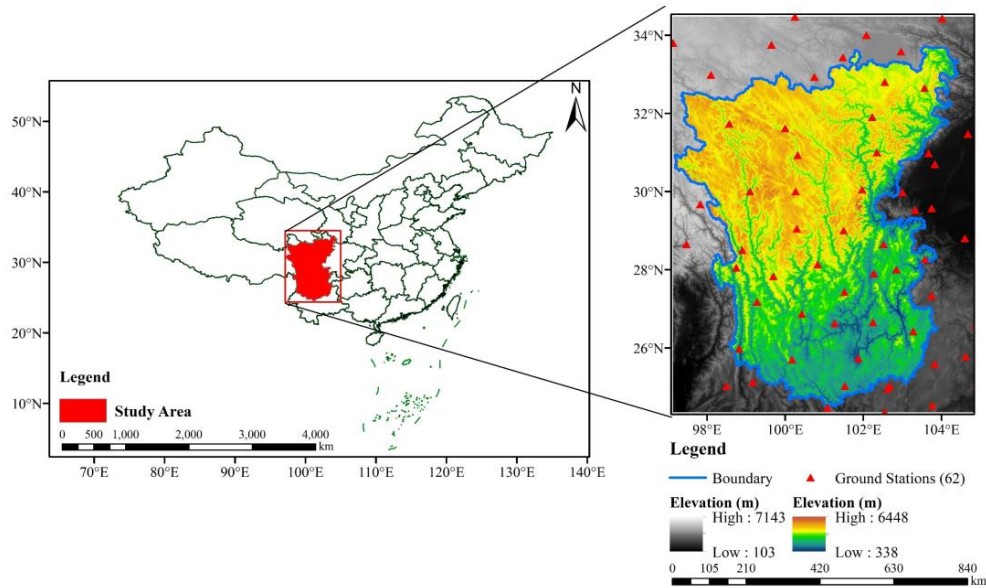

**Figure 1.** Spatial distribution of ground-based stations in and around the HMR.

### 2.2. Data

#### 2.2.1. Ground Observations of Precipitation

Although rain gauge stations in the HMR are sparse, and access to ground-observed precipitation data is quite limited, they are still the most direct way to obtain accurate measurements. Daily ground-observed precipitation datasets for HMR from 2001 to 2016 were obtained from the China Meteorological Administration (CMA; http://data.cma.cn/ accessed on 11 March 2022). The data were subjected to rigorous quality assurance, whereby missing data were supplemented by migrating the adjacent gauge data upon verification. Several representative stations within ~0.5° of the HMR were included as buffers to reduce the inaccuracy caused by sparse rain gauge stations. Sixty-two rain gauge stations with no missing data from 2001 to 2016 were identified as the reference for the intercomparison of the precipitation product datasets. Figure 1 shows the even spatial distribution of ground observation stations, which were located between 97.1°E and 105.0°E and between 24.4°N and 34.5°N. The elevation of the stations ranged from 340 to 4400 m. The wide range of heights facilitates the comparison of vertical distinctions.

2.2.2. Gridded Precipitation Datasets

Three gridded precipitation products were used and compared to the reference dataset, including IMERG, TRMM, and MSWEP. Table 1 summarizes the characteristics of the three gridded precipitation products.

The MSWEP is a global precipitation dataset spanning 1979–2017 (http://www.gloh2o.org/ accessed on 11 March 2022). The latest version of MSWEP has several advantages, including (1) comprehensive global coverage, (2) high spatial and temporal resolution, (3) optimal merging of global data based on various gauges, satellites, and reanalysis estimates, and (4) distributional bias corrections to eliminate spurious light rain and restore attenuated peaks [27]. The TRMM monitored tropical rainfall as a joint mission launched by NASA and JAXA. The TRMM 3B42 product applies the TRMM TMPA algorithm, providing one of the best precipitation estimates from a wide variety of modern satellite-borne sensors [51]. The last version, TRMM-3B42 V7 (Source: https://disc.gsfc.nasa.gov/datasets accessed on 11 March 2022), is a post-real-time product with 0.25° spatial resolution that combines rainfall estimates from PMW and IR sensors, microwave imager, and infrared scanner radiometers [28]. IMERG is a representative product of the GPM mission, which has higher accuracy and spatiotemporal resolution than previous precipitation products (source: https://gpm.nasa.gov/data/directory accessed on 11 March 2022). As a successor of TRMM, the GPM satellite has an advanced radar/radiometer system, including the GPM microwave imager and a dual-frequency precipitation radar [52]. In addition, various passive microwave and infrared satellite sensors support rainfall estimation. According to demand, there are three modes of IMERG: early, late, and final. The final-run dataset was corrected based on the GPCC monthly gauge analysis, which showed better consistency and event detection on both basin and grid scales, whereas the early and late runs showed larger deviations in arid regions [53–55]. The latest version of the IMERG Final Run (IMERG V6), launched in April 2014, was used. All relevant data were converted from the coordinated universal time (UTC) to local standard time (LST).

**Table 1.** Summary of characteristics for the three gridded precipitation datasets.

| Dataset | Period | Coverage | Temporal Resolution | Spatial Resolution |
|---------|--------|----------|---------------------|--------------------|
| GPM-IMERG V6 | 2001–2020 | 60°S–60°N | 0.5 h | 0.1° |
| MSWEP V2.2 | 1979–2017 | Global | 3 h | 0.1° |
| TRMM-3B42 V7 | 1998–2020 | 50°S–50°N | 3 h | 0.25° |

*2.3. Methodology*

2.3.1. Statistical Analysis

Several commonly used continuous statistical metrics were used: correlation coefficient (*CC*), root mean square error (*RMSE*), mean absolute error (*MAE*), and relative bias (*RB*). These metrics were calculated as follows:

$$CC = \frac{\sum_{i=1}^{N} (P_i - \overline{P})(O_i - \overline{O})}{\sqrt{\sum_{i=1}^{N} (P_i - \overline{P})^2 \sum_{i=1}^{N} (O_i - \overline{O})^2}} \tag{1}$$

$$RMSE = \sqrt{\frac{\sum_{i=1}^{N} (P_i - O_i)^2}{N}} \tag{2}$$

$$MAE = \frac{\sum_{i=1}^{N} |P_i - O_i|}{N} \tag{3}$$

$$RB = \frac{\sum_{i=1}^{N}(P_i - O_i)}{\sum_{i=1}^{N} O_i} \qquad (4)$$

where $P_i$ and $O_i$ represent gridded product and ground observation precipitations, with $\overline{P}$ and $\overline{O}$ being their respective mean values, and $N$ is the number of collocated observations.

The $CC$ value describes the degree of correlation in the linear relationship between two variables [56]. A larger $CC$ value indicates a higher agreement between the gridded products and observed precipitation. The ideal $CC$ value is 1. The $RMSE$ and $MAE$ measure the average error and average difference, respectively. $MAE$ is a linear measurement of error, whereas $RMSE$ is a quadratic loss function that points to extremes [57]. The smaller the $RMSE$ and $MAE$, the smaller the error between the gridded products and observed precipitation. The $RB$ quantifies the degree to which the gridded products overestimate or underestimate the observed precipitation. A positive (or negative) value indicates that the datasets overestimate (or underestimate) relative to the observations. The ideal values for the $RMSE$, $MAE$, and $RB$ were 0. $RRMSE$ and $RMAE$ are defined as the $RMSE$ and $MAE$ divided by the average of the gridded precipitation dataset, respectively, which indicates the normalized values of $RMSE$ and $MAE$ for comparison purposes.

Several widely used categorical statistical indices, including the probability of detection ($POD$), false alarm ratio ($FAR$), $BIAS$, and critical success index ($CSI$), which are calculated as follows, were adopted to evaluate the precipitation detection capabilities of the three gridded precipitation products:

$$POD = \frac{H}{H + M} \qquad (5)$$

$$FAR = \frac{F}{H + F} \qquad (6)$$

$$BIAS = \frac{H + F}{H + M} \qquad (7)$$

$$CSI = \frac{H}{H + F + M} \qquad (8)$$

where $H$ is the number of precipitation events detected within the gridded precipitation datasets and gauge monitoring data, $M$ is the number of precipitation events detected within the gauge monitoring station data but not within the gridded precipitation datasets, and $F$ is the number of precipitation events detected within the gridded precipitation datasets but not within the gauge monitoring station data. Since both gauges and gridded precipitation products are prone to large uncertainties for light precipitation, a 1 mm/day value is typically adopted as the threshold for determining rain/no-rain events [58]. The $POD$ represents a "hit rate", which describes the proportion of the observed precipitation events detected correctly by the evaluated product [59]. $FAR$ is the proportion of precipitation events identified by the gridded products that were not observed by the gauge station [60]. $BIAS$ indicates the degree of deviation between the identification of the product and real rainfall events. $CSI$ demonstrates the overall ability of products to detect precipitation occurrence [30]. Values of 1 for $POD$, $BIAS$, and $CSI$ and values of 0 for $FAR$ indicate a relatively ideal product performance.

2.3.2. Extreme Precipitation Analysis

To compare the applicability of precipitation products in characterizing extreme precipitation, 11 widely used extreme precipitation indicators (Table 2) were selected for this study [13,24,61,62].

**Table 2.** Extreme precipitation indices used in this study.

| Index | Descriptive Name | Definition | Units |
|---|---|---|---|
| R0.1mm | Number of precipitation days | Annual count of days when precipitation $\geq$ 0.1 mm | day |
| R10mm | Number of moderate precipitation days | Annual count of days when precipitation $\geq$ 10 mm | day |
| R12mm | Number of erosion precipitation days | Annual count of days when precipitation $\geq$ 12 mm | day |
| R25mm | Number of heavy precipitation days | Annual count of days when precipitation $\geq$ 25 mm | day |
| PRCPTOT | Wet-day precipitation | Sum of daily precipitation > 1.0 mm | mm |
| SDII | Simple daily intensity index | Annual total precipitation divided by the number of wet days | mm/d |
| RX1day | Maximum 1-day precipitation | Maximum 1-day precipitation total | mm |
| CWD | Consecutive wet days | Maximum number of consecutive wet days | day |
| CDD | Consecutive dry days | Maximum number of consecutive dry days | day |
| R95p | Total annual precipitation from very wet day | Annual sum of daily precipitation > 95th percentile | mm |
| R99p | Total annual precipitation from extremely wet day | Annual sum of daily precipitation > 99th percentile | mm |

Notes: A dry (or wet) day is defined as one in which the daily precipitation is lower (or no less) than 1 mm/day.

Extreme precipitation is typically determined by a threshold that can be absolute or relative [63]. The absolute threshold is a fixed value that can seriously affect human production, life, and the ecological environment, taking 50 mm/day as the threshold for a rainstorm [64]. Relative thresholding most commonly uses the percentile value of a data series as the threshold [65]. Because the spatial distribution of precipitation in the study area varies widely, the maximum daily precipitation at some stations did not reach 50 mm. Therefore, defining extreme precipitation events in terms of absolute thresholds would have affected the results. To address this limitation, the daily scale data for each site from 2001–2016 were arranged in descending order, and the 99th percentile value was used as the threshold value, defined as Max99, when the precipitation at the corresponding site exceeded this threshold. Similarly, the 95th, 90th, and 80th percentiles were used as thresholds to define Max95, Max90, and Max80, respectively. The deviations between the product datasets and ground observations were compared using these four daily heavy rainfall sequences. The study area was divided into four altitude gradients based on the mountain terrain classification of geomorphology [66], including high altitude (4000–5000 m), sub-high altitude (2500–4000 m), medium altitude (1000–2500 m), and low altitude (<1000 m), to explore the applicability of the three products at different altitudes.

### 3. Results

*3.1. Comparison of the Applicability of the Three Products in the Study Area*

Figure 2 shows the scatter plots of the three gridded precipitation products versus ground observed precipitation at the watershed scale. Because extreme precipitation events usually occur within a short period, daily timescale was considered. The fits of the gridded precipitation data to the ground observation data for all three products were relatively close. The results of the categorical statistical indices for the three gridded precipitation products are shown in Figure 3. The *POD* results suggested that all products had high identification accuracy (all *POD* values exceeded 0.8), whereas MSWEP was slightly better than IMERG and TRMM. MSWEP had the highest *FAR* value, which was notably lower than that of IMERG. For the *BIAS* results, IMERG and MSWEP were lower and higher, respectively, than the ideal values (value of 1). TRMM registered the best *BIAS* values. The *CSI* values for all three gridded products were close to 0.8, indicating similar overall detection abilities.

The various statistical indicators of the gridded and observed precipitation on a daily scale are presented in Table 3. Most of the statistical indices were well captured by IMERG. The TRMM results were slightly worse than those of IMERG, both with standard deviations similar to the observed precipitation. In contrast, MSWEP results were the worst, with

lower kurtosis, skewness, standard deviation, and coefficient of variation than that of TRMM and IMERG.

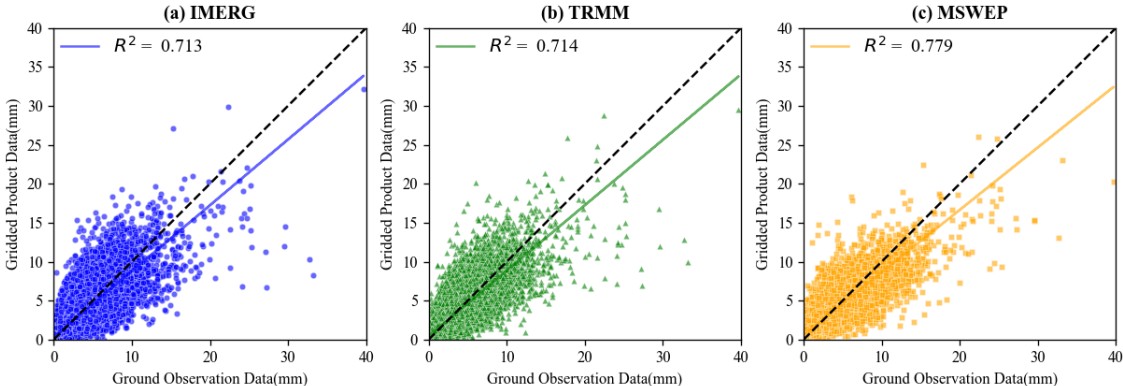

**Figure 2.** Daily precipitation scatterplots of ground observations versus the three products (**a**) IMERG, (**b**) TRMM, and (**c**) MSWEP at the watershed scale from 2001 to 2016. The black dashed line indicates the 1:1 correspondence, and the blue, green, and yellow solid lines represent the best linear regression fits for IMERG, TRMM, and MSWEP, respectively.

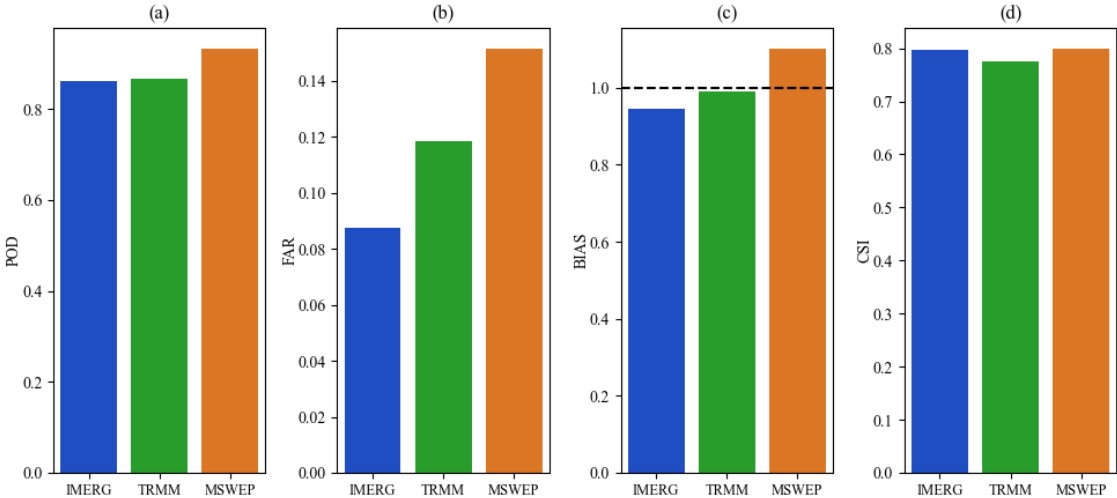

**Figure 3.** Categorical statistical indices ((**a**) *POD*, (**b**) *FAR*, (**c**) *BIAS*, and (**d**) *CSI*) of the three precipitation products at a daily scale.

**Table 3.** Comparison of daily statistical results of the precipitation products.

| Statistical Indices | CMA | IMERG | MSWEP | TRMM |
|---|---|---|---|---|
| Mean (mm) | 2.19 | 2.18 | 2.31 | 2.28 |
| Median (mm) | 1.307 | 1.233 | 1.613 | 1.370 |
| Upper 95th percentile (mm) | 7.949 | 8.088 | 7.556 | 8.221 |
| Upper quartile (mm) | 3.507 | 3.576 | 3.879 | 3.681 |
| Lower quartile (mm) | 0.280 | 0.163 | 0.471 | 0.270 |
| Lower 95th percentile (mm) | 0.000 | 0.005 | 0.092 | 0.011 |
| Range (mm) | [0, 39.657] | [0, 32.129] | [0, 25.975] | [0, 29.557] |
| Skewness | 2.314 | 1.864 | 1.603 | 1.819 |
| Kurtosis | 9.637 | 4.858 | 3.821 | 4.612 |
| Standard deviation (mm) | 2.849 | 2.849 | 2.587 | 2.830 |
| Coefficient of variation (%) | 1.213 | 1.223 | 1.031 | 1.167 |

### 3.2. Comparison of Extreme Precipitation Indices

Table 4 shows the average extreme precipitation indices for the ground observation data and the three gridded products. Figure 4 shows boxplots for the six absolute threshold extreme precipitation indices of the observed data and three products: R0.1, R10, R12, R25, PRCPTOT, and SDII. From Table 4 and Figure 4, it can be seen that all the gridded products overestimated the R0.1mm value. Among all the products datasets, the R0.1mm values for the IMERG product were closest to the value of the observed data, whereas the R0.1mm values for MSWEP were significantly larger, exceeding by 230 days. For the R10 mm and R12mm results, there was no particular difference in the performance of the three precipitation products, and the deviation from the observation was approximately one day. MSWEP slightly underestimated the values for R12mm and severely for R25mm. For the R10, R12, and R25mm results, we found that IMERG and TRMM had stronger detection abilities for moderate precipitation (R10mm) and erosion precipitation (R12mm) events. In the case of the PRCPTOT results, TRMM and MSWEP overestimated the total precipitation on wet days; however, the degree of overestimation for IMERG was lower (the average PRCPTOT value exceeded approximately 18 mm). For SDII, all three gridded products were below the observed value (6.54 mm/day) to various degrees. It should be noted that IMERG was the product closest to the observed data among the three products evaluated.

Figure 5 shows boxplots of the second category of extreme precipitation indices, the maximum indices of precipitation, with respect to the daily precipitation values (RX1day), the duration of wet/dry periods (consecutive wet days (CWD) and consecutive dry days (CDD)), and the percentile indices (R95p and R99p). As seen in Table 4 and Figure 5, for the RX1day indices, it can be seen that IMERG had the closest mean RX1day (60.81 mm) to the observed value (57.10 mm), whereas MSWEP underestimated it. These results were also in line with those for the R95p and R99p indices. For the CWD and CDD indices, IMERG had the smallest deviation in these two indices, with CWD overestimated by ~4 days and CDD underestimated by ~3 days. Upon closer inspection, IMERG and TRMM could capture extreme precipitation events better. Most of the indices were close to the observed data, and both performed consistently for indices reflecting larger intensity precipitation (such as R12mm and R25mm). Compared to MSWEP, IMERG and TRMM performed more accurately for indices reflecting extremely intense precipitation (R95p, R99p, and RX1day). Although MSWEP reflected moderate precipitation days (R10mm = 24 days) well, it severely overestimated the number of precipitation days (R0.1mm = 233 days), CWD (48 days), and the lowest R95p (370.92 mm), R99p (125.60 mm), and R25mm (4 days), which suggests that MSWEP showed a poor ability to identify high-intensity precipitation.

**Table 4.** Comparison of extreme precipitation indices for the observed data and the three products.

| Metric | CMA | IMERG | MSWEP | TRMM |
|---|---|---|---|---|
| R0.1mm (days) | $128 \pm 3$ | $147 \pm 3$ | $233 \pm 4$ | $172 \pm 6$ |
| R10mm (days) | $25 \pm 1$ | $25 \pm 1$ | $24 \pm 1$ | $26 \pm 1$ |
| R12mm (days) | $20 \pm 1$ | $21 \pm 1$ | $18 \pm 1$ | $21 \pm 1$ |
| R25mm (days) | $6 \pm 1$ | $7 \pm 0$ | $4 \pm 0$ | $7 \pm 0$ |
| PRCPTOT | $832.13 \pm 32.55$ | $849.93 \pm 21.17$ | $916.05 \pm 28.85$ | $885.44 \pm 26.09$ |
| SDII (mm/day) | $6.54 \pm 0.21$ | $5.91 \pm 0.22$ | $3.93 \pm 0.10$ | $5.55 \pm 0.26$ |
| RX1day (mm) | $57.10 \pm 3.42$ | $60.81 \pm 2.52$ | $40.75 \pm 2.31$ | $56.35 \pm 2.49$ |
| CWD (days) | $11 \pm 0$ | $15 \pm 1$ | $48 \pm 3$ | $21 \pm 2$ |
| CDD (days) | $39 \pm 1$ | $36 \pm 1$ | $18 \pm 1$ | $30 \pm 1$ |
| R95p (mm) | $457.08 \pm 19.46$ | $478.42 \pm 17.20$ | $370.92 \pm 14.81$ | $463.80 \pm 15.86$ |
| R99p (mm) | $168.55 \pm 8.90$ | $180.33 \pm 7.09$ | $125.60 \pm 6.33$ | $169.75 \pm 6.55$ |

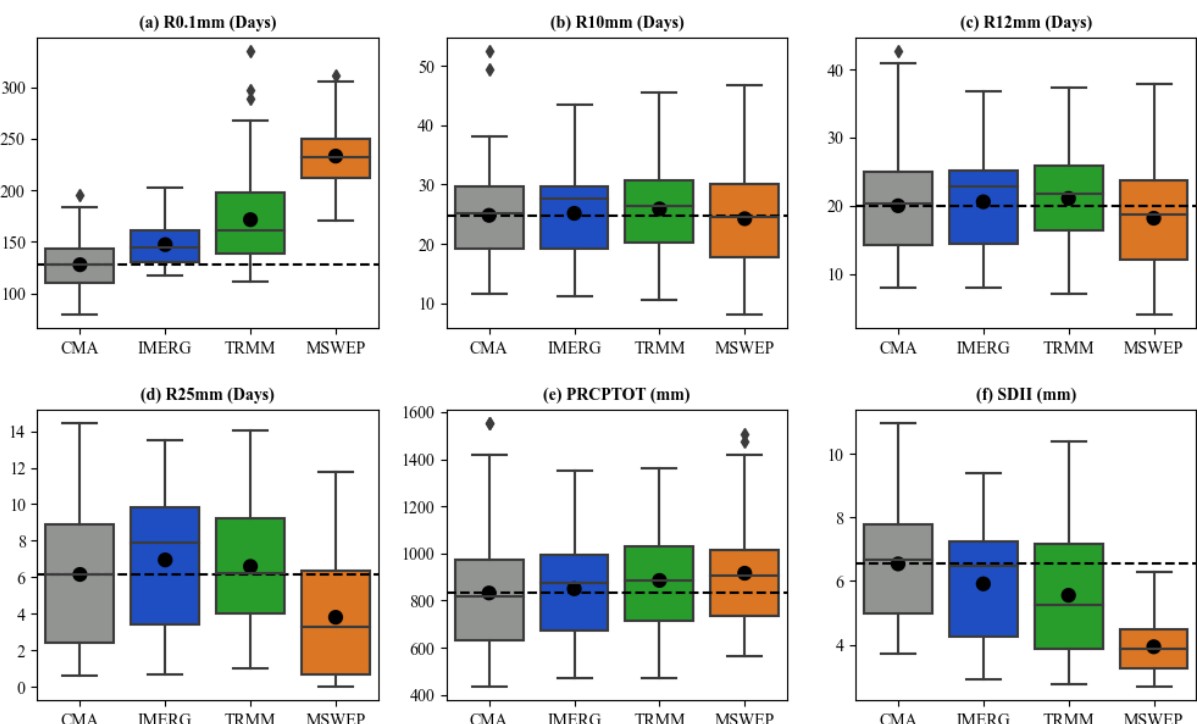

**Figure 4.** Boxplots for the six absolute threshold indices: (**a**) R0.1mm, (**b**) R10mm, (**c**) R12mm, (**d**) R25mm, (**e**) PRCPTOP, and (**f**) SDII for the observed data and the three products. The upper and lower edges of the box mark the upper and lower quartiles (75% and 25%, respectively). The solid line in the box marks the median value. The uppermost and lowermost horizontal lines mark the 1.5I QR outliers. The point marks the average value. The black dashed line indicates the mean value of the indices of the observed data.

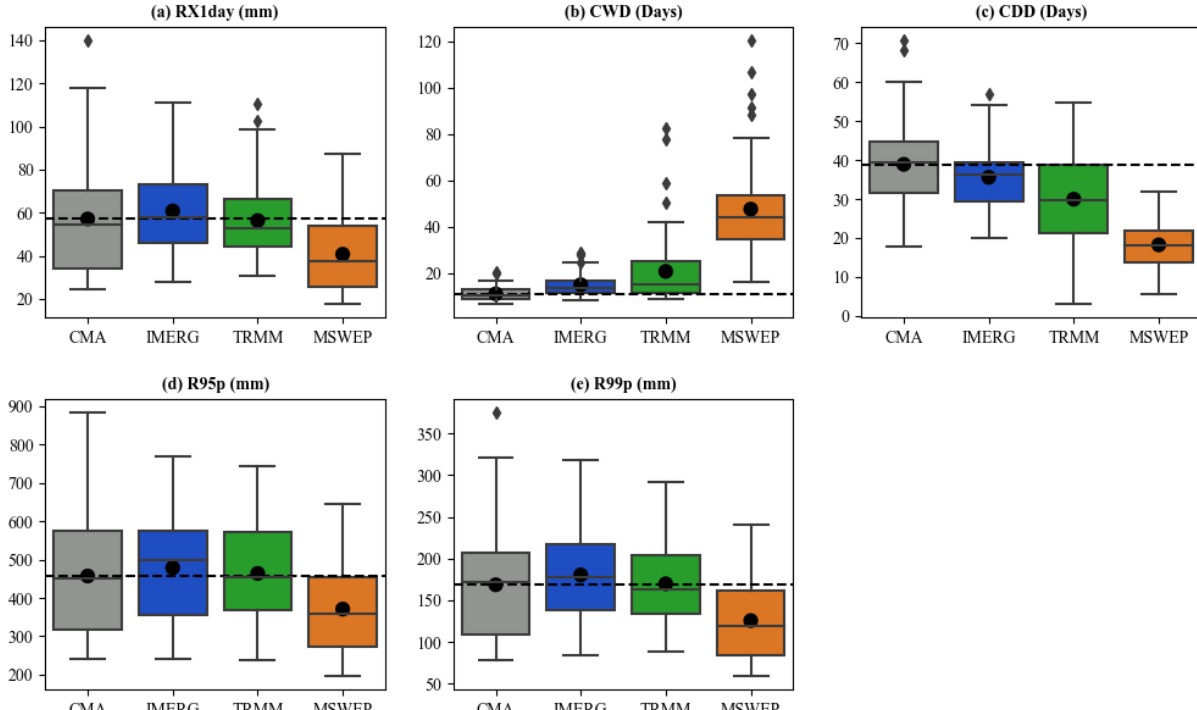

**Figure 5.** Boxplots for the three maximum indices: (**a**) RX1day, (**b**) CWD, and (**c**) CDD, and two percentile indices, (**d**) R95p and (**e**) R99p, for the observed data and the three products.

To examine the temporal agreement between the three gridded products and observed data across the entire study area, annual time series were generated for each dataset. Figure 6 shows the mean extreme precipitation indices for all 62 rain gauge stations in the HMR. The MSWEP deviated the furthest from the observed data for each index, overestimating R0.1mm, PRCPTOT, and CWD, and underestimating R12mm, R25mm, SDII, RX1day, CDD, CWD, R95p, and R99p. Both IMERG and TRMM were close to the observed data for R10mm, R12mm, and R25mm, but underestimated the SDII and overestimated the CWD. The values of the four continuous statistical metrics (CC, RRMSE, RMAE, and RB) for the series of extreme precipitation indices are shown in Figure 7. TRMM had the lowest CC values for R0.1mm, SDII, CWD, and CDD, while MSWEP had the highest for R0.1mm and CDD. For the other extreme precipitation indices, the CC values of the three products were very similar. The RRMSE and RMAE results were similar, with MSWEP showing the largest values for all extreme precipitation indices. The RB values of IMERG were near zero for all indices except CWD, whereas those of MSWEP were the farthest from zero in all three datasets, suggesting that MSWEP had the most considerable bias for extreme precipitation in the HMR.

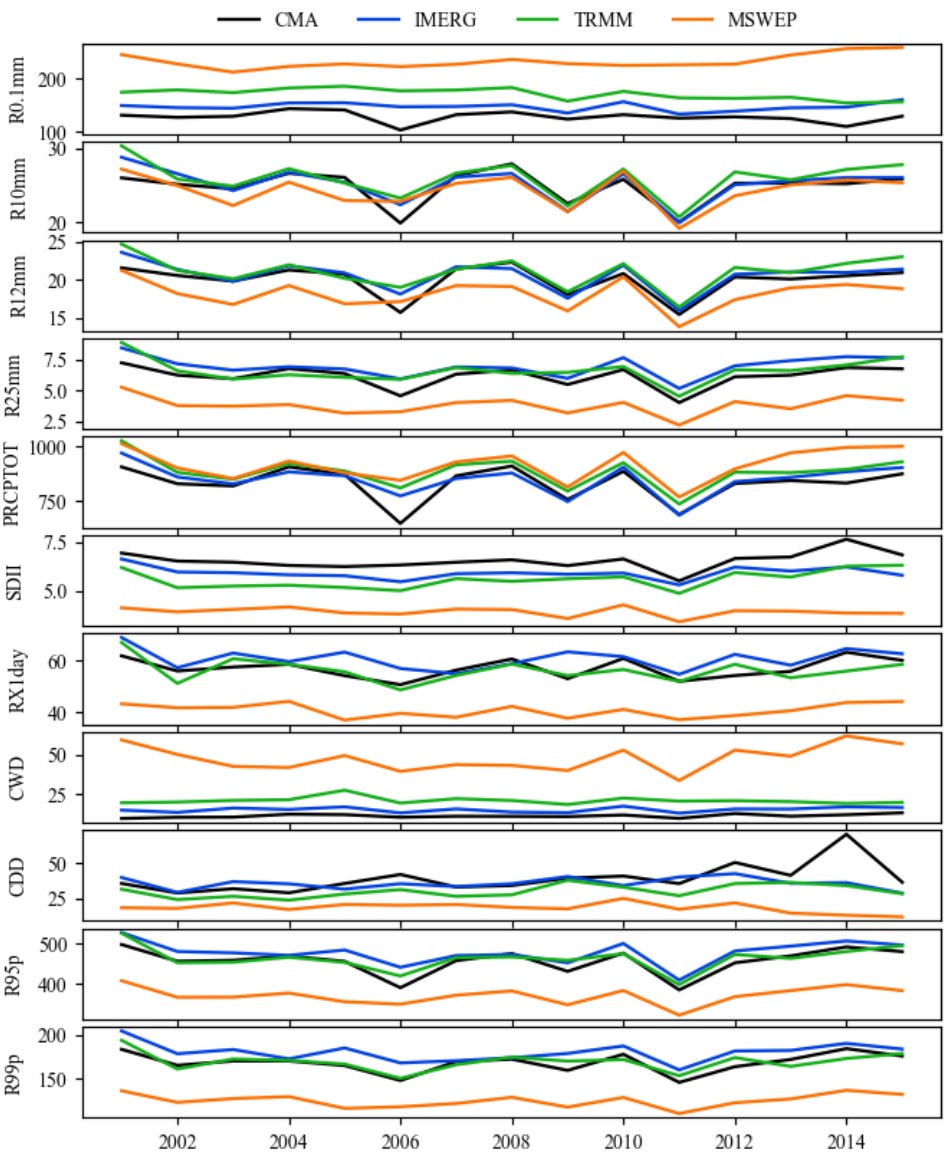

**Figure 6.** Mean extreme precipitation indices of the 62 rain gauge stations, and the corresponding IMERG, MSWEP, and TRMM estimates from 2001 to 2016.

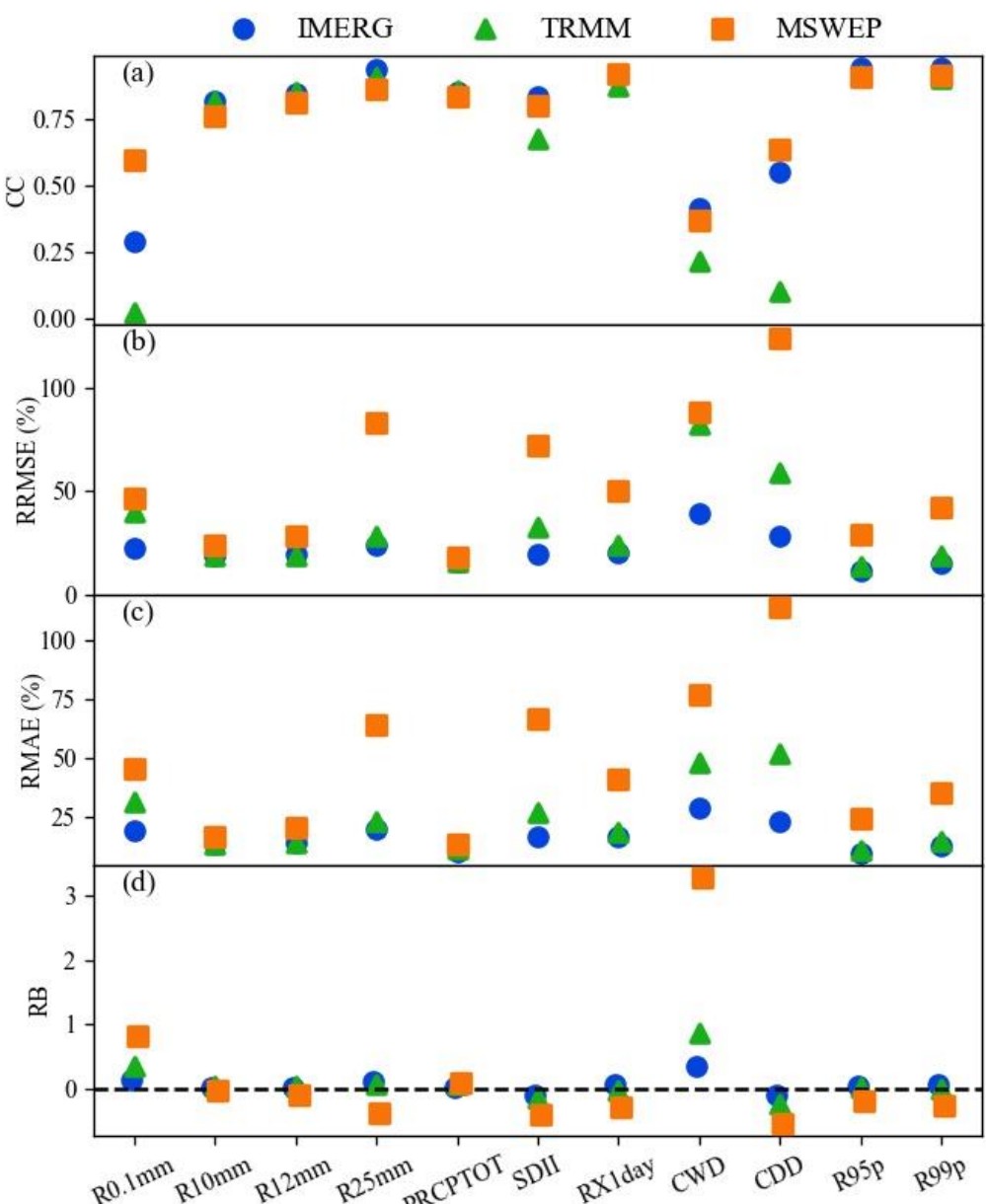

**Figure 7.** Distribution of continuous statistical metrics ((**a**) CC, (**b**) RRMSE, (**c**) RMAE, and (**d**) RB) of the comparisons between the three precipitation products and ground observations at different extreme precipitation indices.

To more clearly compare the recognition ability of the three precipitation products for high-intensity precipitation, the extreme precipitation indicators that reflect intense precipitation were selected to calculate their relative errors at each station separately. Their density distribution curves are presented in Figure 8. As can be seen, the distribution curves for IMERG for R12mm and R25mm were more peaked and tighter, with mean values closer to zero. Especially for CWD, the relative errors of IMERG were concentrated around 0, whereas the curve of MSWEP was flatter. For RX1day, R95p, and R99p, the distribution patterns of IMERG and TRMM were relatively similar, whereas the distribution curve of MSWEP had a significant deviation to the left, indicating an underestimation of the amount of high-intensity precipitation. Overall, MSWEP performed poorly compared with the other products. For most indicators, the ability of IMERG to detect extreme precipitation was less uncertain than that of TRMM.

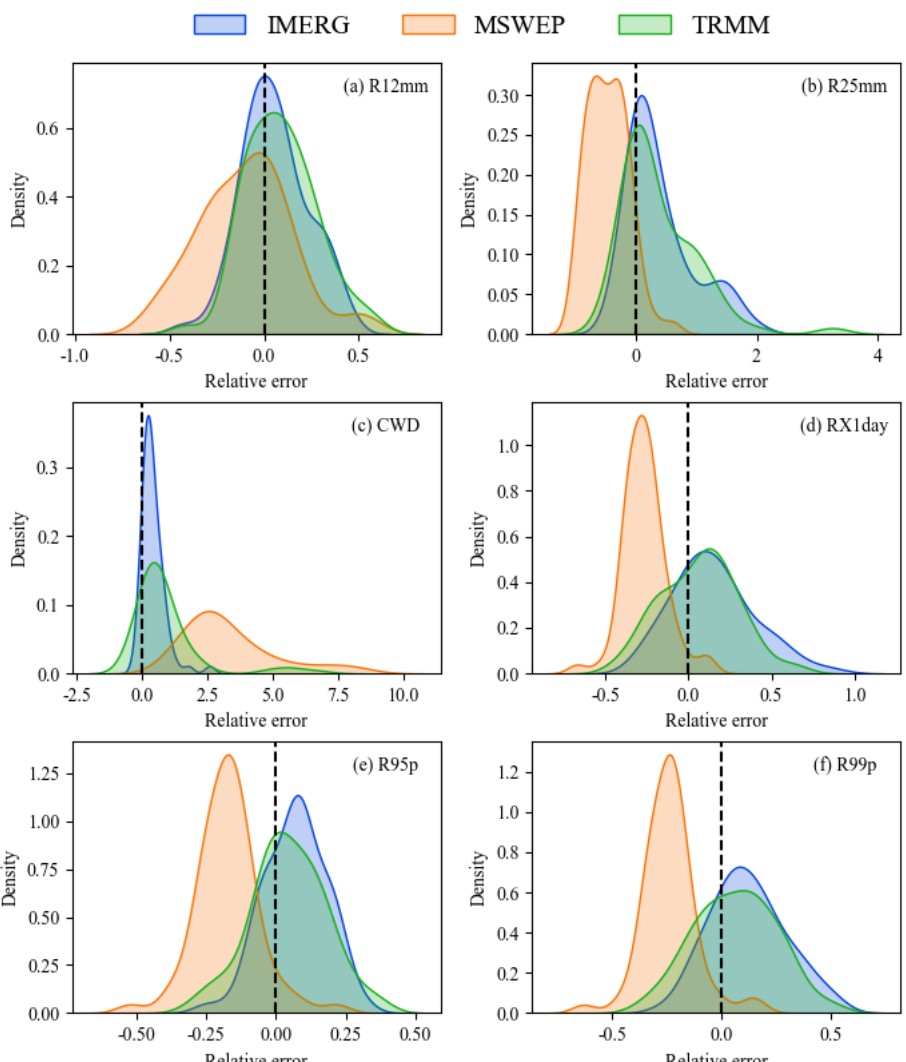

**Figure 8.** Extreme precipitation index: (**a**) R12mm, (**b**) R25mm, (**c**) CWD, (**d**) RX1day, (**e**) R95p, and (**f**) R99p kernel density estimate distribution curves for the three precipitation products.

### 3.3. Comparison of Three Products for Heavy Rainfall Identification

The absolute errors between the ground observation data and gridded products were calculated for the four defined precipitation time series, and the results are shown in Figure 9. All three precipitation products underestimated the actual precipitation to varying degrees at all elevations, with MSWEP underestimating the lowest altitude. The error generated by the three products decreased with increasing rainfall intensity. At Max99, there was a clear trend of decreasing accuracy for the three products as the altitude decreased (Figure 10). The error characteristics of IMERG and TRMM were similar at almost all elevations. For the most part, the deviation of the MSWEP was the maximum, with absolute errors close to −15 at high altitudes and up to −25 at medium altitudes. Figure 11 shows a comparison of the statistical metrics of the Max99. The MSWEP had the largest *CC* value, exceeding 0.6, but also showed the largest deviation (*RRMSE* > 60%, *RMAE* > 25%, *RB* < −0.4). IMERG had the best *RMAE*, *RRMSE*, and *RB* values, indicating a smaller deviation from the observed values. As a result, IMERG had the strongest detection ability for heavy precipitation.

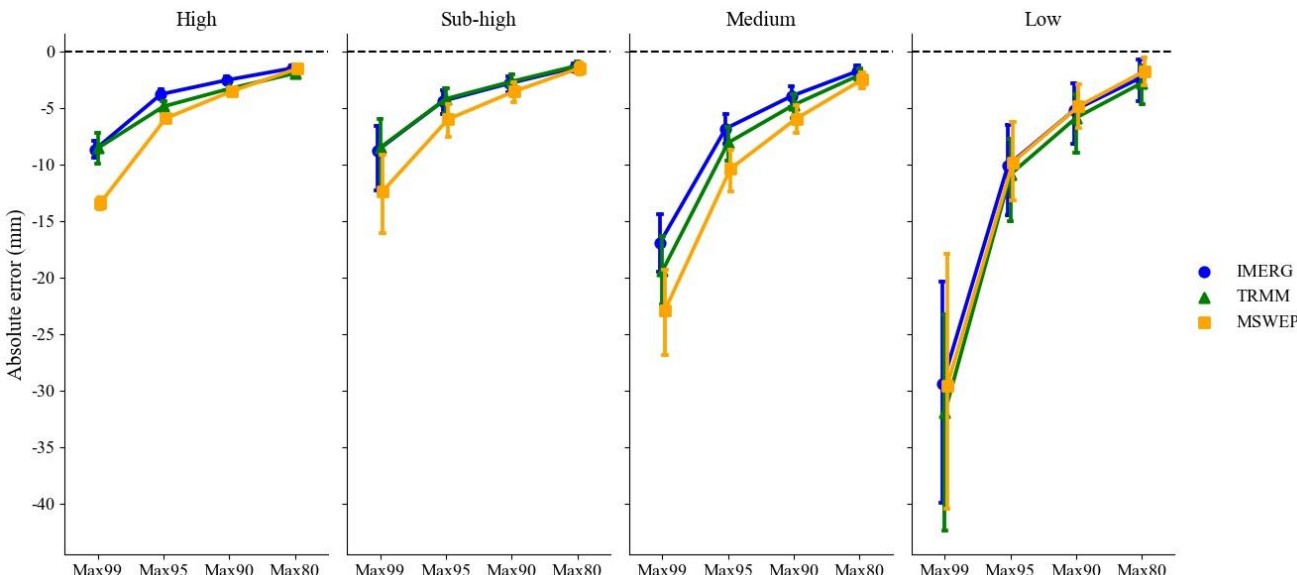

**Figure 9.** Absolute error between precipitation products and observed data on the altitude gradient in the daily heavy rainfall series.

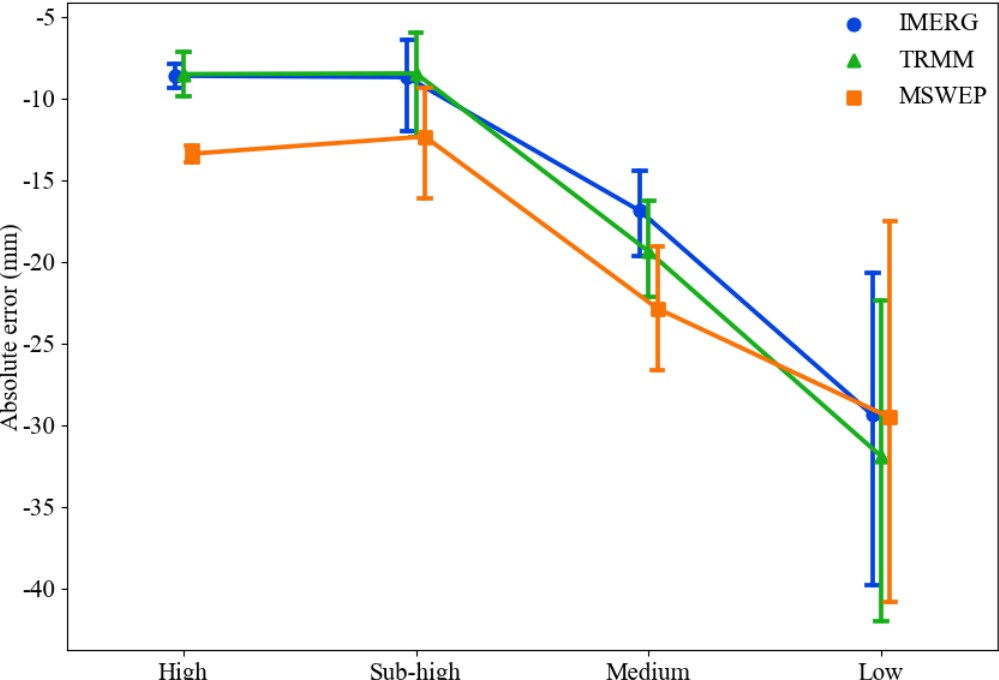

**Figure 10.** Absolute error between precipitation products and observed data on the altitude gradient at Max99.

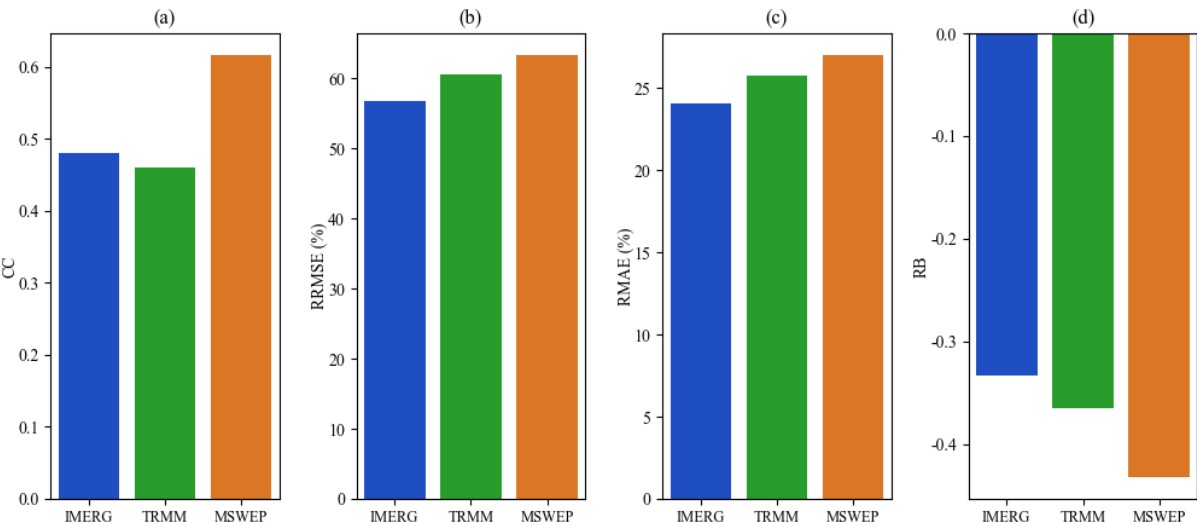

**Figure 11.** Continuous statistical metrics ((**a**) *CC*, (**b**) *RRMSE*, (**c**) *RMAE*, and (**d**) *RB*) of the three precipitation products for Max99.

## 4. Discussion

### 4.1. Effect of Elevation

In this study, the performance of the precipitation products was greatly affected by topographic elevation and showed a consistent trend across the elevation gradient. IMERG exhibits better performance at medium altitudes, which is in line with the findings of previous studies [67]. According to previously published studies [68], the accuracy of precipitation products can be affected by topography and is more accurate in plain areas than in mountainous areas. In mountainous areas with complex terrain, precipitation products may underestimate actual precipitation, possibly because of the low gauge network density in such areas. In addition, precipitation can be falsely detected because of the contrast between the temperature and emissivity of rough surfaces of snow-covered areas [69,70]. Extremely high altitudes and too few stations made the results less reliable at the highest altitudes. Instead, sub-high (2500–4000 m) and medium (1000–2500 m) altitudes are more common in mountainous areas, where the performance of the three precipitation products is better than that at low (<1000 m) altitudes. One of the reasons is that the primary production and living spaces are mainly located at lower elevations, such as in the southeast of the study area, and the living space has gradually increased [49]. As a result of urbanization and land-use changes, the natural surface conditions, hydrological balance, and atmospheric characteristics of a locality are significantly altered [71]. However, most low-altitude stations are distributed in the transitional zone between the Tibetan Plateau and Sichuan Basin. Topographic uplift and latent heat release triggered the development of complex convective systems [72], which decreased the accuracy and sensitivity of satellite-based sensor recognition [32,73,74]. Additionally, because higher-altitude areas are closer to the onboard radar equipment, the effect of radar signal attenuation can be reduced during precipitation data acquisition. Therefore, satellite sensors can more accurately obtain true ground-based observations at higher altitudes. Better proximity to satellites and interaction of convective systems with regional topography may be the primary reasons for the better performance at higher altitudes than that at lower altitudes.

This study and several others found that both IMERG and TRMM underestimate high-intensity precipitation [75–79]. One possible reason is that satellites generally assume that deep clouds cause heavy precipitation, but topographic rain from shallow clouds is ignored. Precipitation products have large errors when estimating topographic precipitation [80]. The above discussion offers a possible explanation for the significant tendency of the detection error to increase with decreasing elevation in Max99 (Figure 10). From a study of the altitude dependency of extreme precipitation trends [81], extreme precipitation

events increased mainly at lower altitudes (1000–1500 m). Accordingly, this study focused on performance at medium altitudes. IMERG is the most adapted product for studying extreme precipitation at this altitude range.

### 4.2. Benefits and Limitations

Heavy rainfall is a key concern owing to the unique climatic conditions and hazard characteristics of the HMR, mainly reflected in the R25mm, RX1day, and R99 indicators. For all three indicators, MSWEP underestimated precipitation to varying degrees. This finding is consistent with that of Nair [82]. Meanwhile, MSWEP also overestimated the index characterizing light precipitation. Especially for R0.1mm, the number of overestimated days was as high as 100 days (see Table 4). The comparison of extreme precipitation indices suggests that the MSWEP does not accurately identify either heavy precipitation or drought, as can also be seen from the overestimated CWD and underestimated CDD. Based on the above analysis, we suggest that using MSWEP to represent extreme precipitation at the HMR may not be appropriate, despite its good correlation with the observed precipitation.

IMERG has a higher spatial and temporal resolution than TRMM and therefore displays more detailed spatiotemporal information [76,83]. Higher spatial resolution improves the accuracy of capturing precipitation events at small spatial scales, making it more suitable for smaller watersheds [75]. The higher temporal resolution makes IMERG more capable of identifying heavy precipitation events, as high-intensity precipitation usually occurs over a short period [78]. In TRMM, precipitation events are often missed in some cases, resulting in a lower CC value. Compared to TRMM, IMERG has more advanced sensors [84,85], which add four channels ranging from 10 GHz to 183 GHz [86]. Augmentation of additional Ka-band frequency radars has undoubtedly improved the performance of IMERG for light/heavy precipitation detection [87]. Our findings are consistent with those of the previous studies that agree that IMERG is more accurate than TRMM in identifying extreme precipitation [67,77,88]. Given the complex topography and climatic conditions of the mountainous regions of Western China, IMERG may help represent precipitation, hazard prevention, and disaster risk reduction. To understand extreme precipitation, IMERG products have the potential to overcome the limitations of insufficient precipitation observations in mountainous areas, assess precipitation trends under continuously changing climatic conditions, evaluate the ecological impacts of extreme weather, and bolster hydrologic modeling with high spatiotemporal resolutions.

However, TRMM and IMERG underestimated high-intensity precipitation and slightly overestimated several extreme precipitation indices (such as R25mm and R99) for ground stations. This suggests that both precipitation products might fail to identify moderate rainfall events observed by gauges as extreme precipitation events but underestimate heavy precipitation. The accuracy of these three gridded products in detecting heavy precipitation varied over the four precipitation sequences (Figure 10). The difference in accuracy was the most pronounced between Max99 and Max95. In fact, the detection skill of all current satellite precipitation datasets decreases as the precipitation thresholds (i.e., 80th to 99th percentiles) increase [89]. The common reasons are that gridded precipitation is homogenized data and high-intensity rainfall attenuates radar signals [78,90]. The differences in the performance of the precipitation products at various precipitation intensities must be investigated in greater detail. In this study, the comparison of the three products in HMR suggests that MSWEP has the worst performance. At the same time, MSWEP has great potential for drought monitoring at larger spatial scales (over mainland China) [35] and exhibits highest accuracy on a larger timescale (annual and monthly scales) among the three products [36,91]. Thus, further research should be undertaken to investigate the reasons for the varied performances of gridded precipitation datasets under different climates, regions, and spatiotemporal scales. Identifying the sources of systematic and random errors enhances satellite sensor systems further [92]. Precipitation patterns worldwide have changed throughout the 20th century [93,94]. It is more challenging to predict and understand extreme precipitation. Despite these uncertainties, the findings

of the present study are valuable. The results of this study can serve as a reference for research on hazard assessment of hydrometeorological disasters and risk management of extreme precipitation.

## 5. Conclusions

We adopted multiple indices to evaluate the performance of the gridded precipitation product datasets by comparing them to the observations from 62 rain gauge stations throughout the study region. The major conclusions can be drawn as follows: MSWEP has the best agreement of daily precipitation time series, but there is a serious underestimation of extreme precipitation. The MSWEP dataset is not sufficiently capable of studying extreme precipitation. IMERG and TRMM have similar detection capabilities for extreme precipitation, both being better than MSWEP, and IMERG provides higher precision and less uncertainty owing to its higher spatiotemporal resolution. The accuracy of these three products in detecting extreme precipitation tended to decrease with decreasing altitude. All three products showed significant errors in identifying heavy rainfall in low-altitude regions in the HMR. Furthermore, the performance of the MSWEP is the worst, except at low altitudes. This study provides a basis for precipitation hazard assessments that require the application of gridded precipitation products.

This study demonstrated that IMERG had the best application potential for risk analyses of extreme precipitation events over the HMR. Given the spatiotemporal resolution of TRMM and its poor performance, in future relevant studies involving different spatial scales, a feasible option would be to use IMERG and TRMM in conjunction.

**Author Contributions:** Conceptualization, G.W. and X.S.; methodology, X.S.; software, J.S.; validation, G.W., X.S. and L.G.; formal analysis, W.D.; investigation, W.D.; resources, W.D.; data curation, W.D.; writing—original draft preparation, W.D.; writing—review and editing, W.D.; visualization, X.S.; supervision, X.S.; project administration, X.S.; funding acquisition, X.S. All authors have read and agreed to the published version of the manuscript.

**Funding:** This study was financially supported by the National Natural Science Foundation of China (41890821, 41790431, and 41977176) and the Strategy Project of the Chinese Academy of Sciences (XDA23090201).

**Data Availability Statement:** The data and codes used for this study are available from the corresponding author upon request.

**Conflicts of Interest:** The authors declare no conflict of interest.

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
