# Peer review of "Evaluation of Three Gridded Precipitation Products in Characterizing Extreme Precipitation over the Hengduan Mountains Region in China"

_remotesensing, doi:10.3390/rs14174408_

Round 1

Reviewer 1 Report

This study is interesting and meaningful for assessing precipitation products in Hengduan Mountains region in China using statistical and extreme precipitation metrics. However, the manuscript is poorly organized, especially for the Introduction section. My main comments are some suggestions for help authors to improve their research. For example, I suggest that the Introduction section should be reorganized and some categorical metrics, which are extremely important for extreme precipitation, should be used in assessing precipitation products. My main comments are as follows:

Introduction section is poorly organized. I suggest authors re-organize it. A good introduction should focus on the problems you are trying to solve, how previous studies has been done on this problem, the inadequacy of previous research and your innovation.

I suggest adding some categorical metrics for assessing precipitation products such as critical success index (CSI), success ratio (SR), probability of detection (POD) and false alarm ratio (FAR). Because they are extremely important at short time scales and for extreme precipitation. These methods were described in Tang et al., (2020).

Tang, G., Clark, M. P., Papalexiou, S. M., Ma, Z., & Hong, Y. (2020). Have satellite precipitation products improved over last two decades? A comprehensive comparison of GPM IMERG with nine satellite and reanalysis datasets. Remote sensing of environment, 240, 111697.

There are too many qualitative expressions in Results section. I suggest adding more data to these qualitative sentences. Such as L211-217.

L42-43. References?

L30-45. You should focus on the importance of this research. That is, why this study is important or necessary. I suggest that authors should point out the problems of previous studies straight to the point in the first paragraph of Introduction. May be this paragraph can be combined with L46-60. In addition, I suggest you focus on why assessing these precipitation datasets in your study area is important, since the lack of in-situ observations is not the reason for this study.

L45-45. This is a general point and may not be important to this study. Please modify it.

TRMM is the name of the satellite, whereas the algorithm is TMPA. Please fix throughout the manuscript. In addition, is including TMPA 3B43 really necessary (since TRMM has been dismissed for quite a while now)?

L61-63. References

L63-71. These examples should relate to the sentence in L61-63. That is, they should relate to “in areas where ground-based precipitation monitoring is limited by comparing them with rain gauge observations.”. More methodology of how previous studies solve the problem related to L61-63 (or you innovation, which is not clearly expressed in the first paragraph of Introduction) should be described. For example, Luo et al. (2021) proposed a new method for assessing precipitation products in ungauged basins using water budget closure…... I know you want to show why these three products were selected in this study. They should be described in following paragraphs.

Luo, Z., Shao, Q., Wan, W., Li, H., Chen, X., Zhu, S., & Ding, X. (2021). A new method for assessing satellite-based hydrological data products using water budget closure. Journal of Hydrology, 594, 125927.

L80-82. May be this sentence need to be double-checked.

L82-84. Double-check and this is your innovation?

L89-91. Good expression. You should bring this sentence forward as I suggested above, since this is one of reasons why you assess the performance of precipitation products in your study area.

L96-98. Maybe this expression is not appropriate (exaggerated the significance of the research).

L123-125. How observed values are processed in this study for keeping their quality? Please detail. In addition, there are only about 30 stations located within the study area rather than 62.

L168-179. Please supplement the value ranges and standards of these indicators for assessing P products.

L191-198. To discuss the error caused by this.

L198-200. Please explain in detail how authors to do this. I compare more about how the authors evaluate precipitation products using extreme precipitation indicators.

Fig. 2. Why there are so many data. Please give the time period of data and the temporal scale in Fig. 2 (2001-2016, daily?).

L210-211. Kriging interpolation may be not necessary. The comparison at rain-gauge stations is sufficient.

L319-322. These look more like method descriptions.

L325-334. Please describe them quantitatively. All qualitative expression in results section should be quantified.

Reviewer 2 Report

Congratulations for this interesting study.

It offers a good tool to represent precipitation, hazard prevention and disaster risk reduction in HMR of China.

I suggest that in the Figure 2 Scatterplots of daily ground ... include daily even that in a previous paragrah is explained.

Author Response

Point : I suggest that in the Figure 2 Scatterplots of daily ground ... include daily even that in a previous paragrah is explained.

Response : Thank you for your suggestions. We have revised it according to your suggestions

Reviewer 3 Report

This work is devoted to comparison of data from three gridded precipitation product to natural data from 62 observation stations for 2001-2016 from Hengduan Mountains region in China. The performance of these precipitation products differing by both spatial and temporal resolution were evaluated with the use of various indices (more than ten indices). All products tested demonstrated more accurate estimation for higher altitudes regions. All of them work with lesser accuracy for low altitudes. The general recommendation is to use these products with caution for the analysis of heavy precipitation in lowland conditions. According to the data of this study it is impossible to say that one of three system used was better fit for analysis than two other. So, the authors conclusion is quite right.

According to mine, as a reviewer, opinion it is better to avoid abbreviations in the papers addressed to a broad scientific auditorium, while in the manuscript under review the use of abbreviations is rather multiple, even in abstract and key-words. My recommendation – to eliminate abbreviations in the abstract and key words and include a list of abbreviations in the end of paper or between abstract and introduction.

Author Response

Point : My recommendation – to eliminate abbreviations in the abstract and key words and include a list of abbreviations in the end of paper or between abstract and introduction.

Response : Thank you for your constructive comments. We have revised it according to your suggestions.

Round 2

Reviewer 1 Report

The manuscript has been greatly improved. All my questions have been answered. Thanks.